# Integrative analysis of gene expression and alternative splicing in microalgae grown under heterotrophic condition

**Bahman Panahi**[1]*, **Mohammad Amin Hejazi**[2]

**1** Department of Genomics, Branch for Northwest & West region, Agricultural Biotechnology Research Institute of Iran (ABRII), Agricultural Research, Education and Extension Organization (AREEO), Tabriz, Iran, **2** Department of Food Biotechnology, Branch for Northwest & West region, Agricultural Biotechnology Research Institute of Iran (ABRII), Agricultural Research, Education and Extension Organization (AREEO), Tabriz, Iran

* panahibahman@ymail.com, b.panahi@abrii.ac.ir

**Data Availability Statement:** All data are available from the ENA database (https://www.ebi.ac.uk/ena) with PRJNA289168 and PRJNA484804 ID.

## Abstract

Heterotrophic cultures are the most effective approach to overcome low growth rate challenge in the most commercial microalgae. However, the mechanism through which heterotrophic condition regulates algae metabolism are not completely clear. Alternative Splicing (AS) is a common posttranscriptional process by which transcriptome and proteome plasticity increases at different environmental conditions. To identify and characterize of AS events in *Auxenochlorella protothecoides* microalga grown in autotrophic and heterotrophic, RNA-Seq data were analysed. We found that AS increased with the transition from autotrophic to heterotrophic condition. 705 and 660 differentially expressed (DEG) and spliced (DAS) genes were identified for *A.protothecoides* was transferred from autotrophic to heterotrophic condition, respectively. Moreover, there was slight coverage between DEG and DAS genes. Furthermore, functional analysis showed that the DAS genes are most frequently related to ion binding and stimulus response. The results also indicated that prevalence of Intron retention is associated with down-regulation of the genes involved in carotenoid biosynthesis. This study provides valuable insights into transcriptional and posttranscriptional plasticity of microalgae during growth mode change.

## Introduction

Heterotrophic cultures are the best alternatives to overcome low growth rates challenge in the most commercial microalgae [1]. In heterotrophic growth mode, organic substrate as sole carbon and nitrogen source are used [2]. Low operating cost and simplicity are the primary advantages of the heterotrophic cultivation. Moreover, heterotrophic cultivation provides an effective approach to produce different metabolites in industrial scale [3].

Although the biological responses of microalgae in heterotrophic condition have been surveyed, the underlying molecular mechanisms are not well known. It has been reported that heterotrophic growth mode affects carbon and nitrogen metabolisms [4]. Moreover,

**Funding:** The author(s) received no financial support for the research, authorship, and/or publication of this article.

**Competing interests:** The authors have declared that no competing interests exist.

photosynthesis, carotenoids production, respiration, as well as protein synthesis are affected by changing of growth mode [5, 6].

Alternative Splicing (AS) serves a transcendental role in increasing coding capacity [7]. The functional impacts of AS fall into two main categories: protein-level alterations and transcript-level modifications [8]. It has been found that AS can act as a regulatory mechanism during developmental processes or in response to environmental conditions [9]. In addition, changing environmental condition has been shown to influence the splicing pattern [10]. Plants even seem to regulate their transcriptome post-transcriptionally in response to quickly changing environmental conditions by using alternative splicing mechanisms [11].

Like in higher plants, AS has a fundamental role in increasing biological complexity in algae. Prior study based on expression sequence tags (EST) analysis indicates that about 2.9% of all genes in *Volvox carteri* undergo alternative splicing [12], which is much lower than higher plants [8]. Identification of AS events based on ESTs analysis is mainly depending to the ESTs number, it is likely that with the advent of high throughput next generation sequencing (NGS) technology and development of new analytical pipelines more details about functional impacts of AS events will be deciphered [13].

Despite NGS technology has been widely applied to dissect of AS change during environmental and nutritional stresses in plant species [9; 14], it remains largely unknown whether or how AS is involved in microalgae responses to different growth condition.

To investigate the changes of AS during transition from autotrophic to heterotrophic growth condition, a set of throughput RNA sequencing data of *Auxenochlorella protothecoides* microalga growing at autotrophic and heterotrophic condition was retrieved from the European Nucleotide Archive (ENA) database. Then, differentially expressed (DEGs) and spliced (DAS) genes between autotrophic and heterotrophic conditions (at five time points, including hour 3, hour 6, hour 12, hour 24, and hour 72) were compared. DAS genes were mostly enriched in cellular metabolic process, nitrogen compound metabolic process, and biosynthetic process. Results of current study also demonstrated that independent mechanism of gene regulation implies by AS during transition to the heterotrophic growth condition.

## Methods and material

### Transcriptome data of *Auxenochlorella protothecoides*

RNA-seq data of *Auxenochlorella protothecoides* which were collected at heterotrophic growth mode (accession number: PRJNA484804) [4], were retrieved from the European Nucleotide Archive (ENA). The data contained raw paired ends data of cells grown at autotrophic growth condition (AG) and heterotrophic growth condition (HG). In AG, *A. protothecoides* cells were cultured in a medium containing 0.7 g $KH_2PO_4$, 0.3 g $K_2HPO_4$, 0.3 g $MgSO_4.7H_2O$, 0.3 mg $FeSO_4.7H_2O$, 0.01 mg vitamin B1, and 1 mL A5 trace mineral solution adding 5 g/L glycine under illumination of 2000 lux. The cells were grown in 100 mL flasks on a shaker set at 220 rpm and 28 ±1˚C [4]. In HG, cells were grown in a medium containing 0.7 g $KH_2PO_4$, 0.3 g $K_2HPO_4$, 0.3 g $MgSO_4.7H_2O$, 0.3 mg $FeSO_4.7H_2O$, 0.01 mg vitamin B1, and 1 mL A5 trace mineral solution adding 30 g/L glucose and 0.5 g/L glycine under dark condition.

Two biological and technical replicates were used for both autotrophic and heterotrophic condition and each corresponding time points after transition to heterotrophic condition (autotrophic condition: SRR7650945 and SRR7650944; 3 hour of growth in heterotrophic culture: SRR7650947, SRR7650946; 6 hour of growth in heterotrophic culture: SRR7650952, SRR7650943; 12 hour of growth in heterotrophic culture: SRR765095, SRR7650950; 24 hour of growth in heterotrophic culture: SRR7650942, SRR7650941; 72 hour of growth in heterotrophic culture: SRR7650949, SRR7650948).

## Processing of RNA-seq data

The quality control of raw data was performed using FastQC software version v.0.11.5. Low-quality reads were filtered out using Trimmomatic v.0.36 [15] with the following parameters [LEADING: 30, TRAILING: 3, SLIDINGWINDOW: 4:20, and MINLEN: 30]. Low-quality bases (quality scores <3) were trimmed out. Trimmed reads were mapped to the *A. prototothecoides* genome (assembly ASM73321v1) using TopHat2 software version 2.0.12 [16] with following parameters [–read-mismatches 5–read-gap-length 3–read-edit-dist 5–library-type = fr-firststrand–splice-mismatches 0]. Then, sorted Alignment/Map (BAM) files were subjected to Picard software version 2.5.0 (https://broadinstitute.github.io/picard/) to remove potential duplicates.

## Analysis of differentially expressed genes

HTSeq-count [17] was used to quantify gene expression at each sample. Prior study have corroborated that negative binomial based methods are more efficient to identify DEGs in experiments with less biological replicates [18]. Therefore, created count matrix was subjected to negative binomial based methods implemented in the Bioconductor DESeq2 package version 1.10.1 [19]. To avoid biases in fold change, low read counts (<5) across all samples were filtered out. Normalization of count data for all samples was performed using the default settings in DESeq2. Wald's test was applied to determine the log2-fold change between AG and HG condition in five time points viz, 3h, 6h, 12h, 24h, and 72h. Significant differential expression was determined as a fold change $\geq |2|$ with a false discovery rate (FDR) corrected p-value cut off of $\leq 0.05$ [20]. Heat map analysis was also performed to visualize of common DEGs at different time points grown in heterotrophic condition.

## Genome-wide analysis of alternative splicing

Processed RNA-seq data were also used to genome wide study of alternative splicing changes of microalgae at heterotrophic condition. Due to reduced illative power, analysis of alternative splicing using the transcript assembly is challenging, therefore, differential exon usage as evidence for alternative splicing was analyzed by using the state-of-the-art software DEXSeq version 1.16.10 [17]. Exon expression was quantified by using a modified HTSeq-count phyton script that was provided in the DEXSeq package [17].

To test for differential expression usage in each gene across all the samples during the transition from autotrophic to heterotrophic condition following interaction model (~ sample + exon + condition:exon) and a likelihood ratio test were applied. Significant DEU was considered as a false discovery rate (FDR) corrected p-value $\leq 0.05$ [20].

ASTALAVISTA program [21] was also applied to detect AS landscape, including Intron retention (IntronR), alternative 3' splice acceptor (AltA), alternative 5' splice donor (AltD), Exon skipping (ExonS) and complex event. The output from landscape analysis can be found in the S1 and S2 Tables for heterotrophic and autotrophic condition, respectively.

## Functional analysis

To examine the functional impacts of the DEG and DAS genes, GO analysis was performed to screen for the enriched GO terms with Fisher's exact test using the Web Gene Ontology Annotation Plot (WEGO) (http://wego.genomics.org.cn/) [22]. Enriched GO terms were categorized in three Biological Process (BP), Molecular Function (MF), and Cellular Component (CC) group under p-value < 0.05.

## Carotenoids biosynthesis-related AS genes

To identify carotenoids biosynthesis-related genes of *A. protothecoides*, corresponding genes from *Chlamydomonas reinhardtii* and *Chlorella vulgaris* were downloaded from the uniprot database (https://www.uniprot.org/). Then, BLASTP was used to align protein sequences of the *A. prototheoides* (available at https://www.ncbi.nlm.nih.gov/genome/) as reference and the downloaded sequences as the query with cutoff *E* value 1E–5.

## Validation of alternative splicing

Additional independent *A. prototheoides* RNA-Seq dataset PRJNA289168 based on the same conditions as the training set was downloaded from ENA database and used as a test set to verify existence and patterns of splicing in candidate DAS genes at heterotrophic condition. This dataset included three autotrophic samples grown in Bristol's salts plus 0.1% (w/v) proteose peptone and nine samples grown in papaya-based heterotrophic cultures (Bristol's salts plus 12.5% (v/v) Rainbow papaya juice.

# Results

## Dynamic transcriptome changes at heterotrophic growth mode

In total, 3614, 2739, 2869, 3542, and 2234 DEGs were found on hour 3, hour 6, hour 12, hour 24, and hour 72, respectively (Fig 1A). Among the mentioned DEGs, 705 of which were shared in all five time points (Fig 1B). About 54% (hour 3: 45%, hour 6: 58%, hour 12: 47%, hour 24: 65%, hour 72.56%) DEGs were up regulated in HG cells (Fig 1C).

Top up- and down-regulated genes and their function annotations on hour 3, hour 6, hour 12, hour 24, and hour 72 of growth in heterotrophic condition are listed in S3 Table. Results showed that ammonium transporter 1 member 3 (AMT1-3) and purple acid phosphatase 15 (PAP15) are among the top up-regulated and superoxide dismutase 2 (SOD2) is among the top down-regulated genes.

## Genome-wide analysis of alternative splicing in heterotrophic condition

In current study, besides differential expression analysis, genome-wide survey of AS process in *A. prototheoides* was also performed. 5229 AS events in autotrophic condition and 11618 AS events in heterotrophic condition were detected, which indicated that heterotrophic condition dramatically induce the AS process in *A. prototheoides* (S4 Table). The overall statistics of the shared/unique DAS at five time points in comparison with DEGs are shown in Fig 2. 660 DAS were common at different time points (Fig 2). Five types of AS events viz, IntronR, AltA, AltD, ExonS, and others, were considered in current study (Fig 2). Results showed that IntronR and AltD are the most prevalent type of AS in heterotrophic and autotrophic growth condition (Fig 2). On average, 28% of the AS events were IntronR (28%), followed by others (27%), AltD (24%), and AltA (15%), with ExonS (5%) being the least frequent type of AS event at heterotrophic growth mode (Fig 2). Validation of occurrence and pattern of AS process for candidate DAS genes were performed by using another RNA-Seq dataset in heterotrophic and autotrophic growth mode for *A. prototheoides*. AS events analysis in additional datasets detected the similar rate and pattern for AS underwent genes during transition from autotrophic to heterotrophic condition.

## Functional impact of DEG and DAS genes

GO annotation showed that DEG and DAS gene sets had similar GO term coverage indicated that these two gene sets participated in the same biological process and cooperated with each

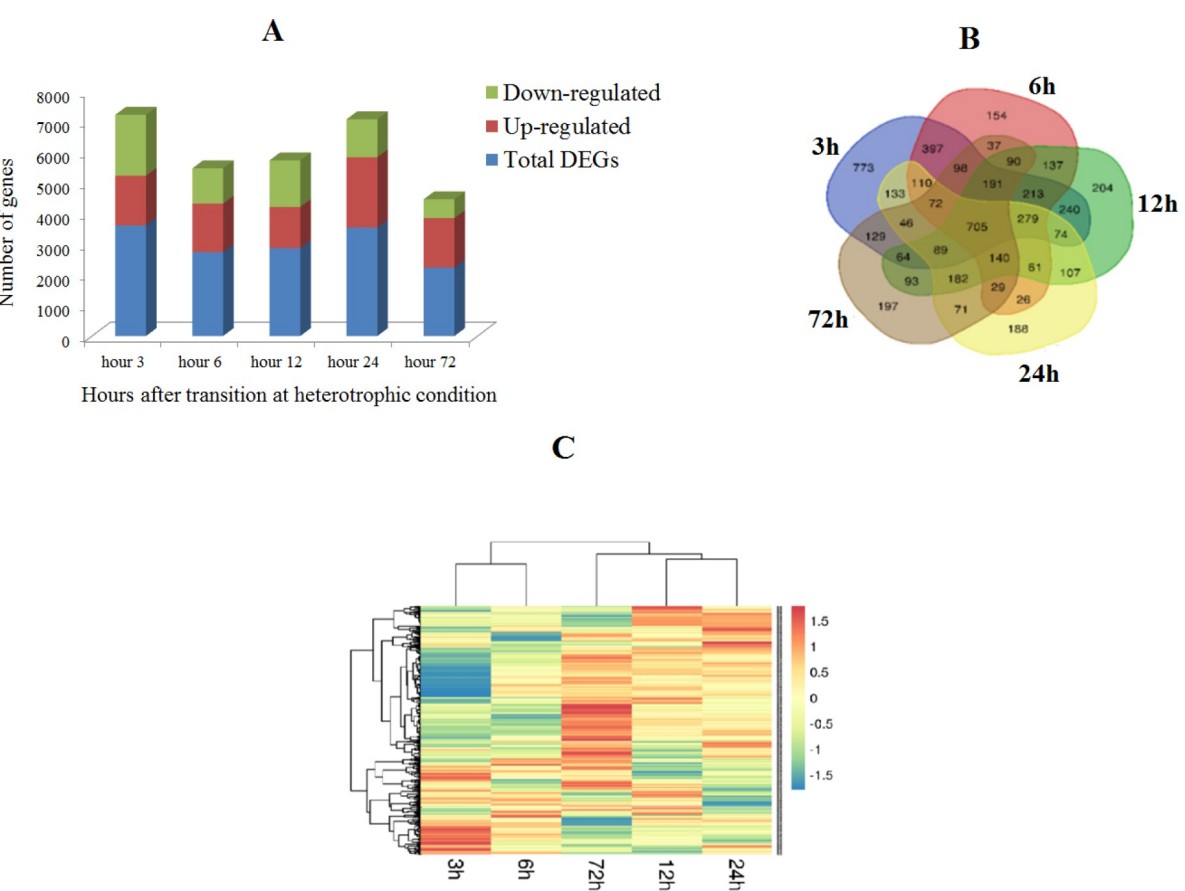

**Fig 1.** Profiles of DEGs in microalgae after transition to heterotrophic condition, frequency of up regulated and down regulated gene (A), overlap of DEGs (B) and expression pattern (C) of common DEGs at five time point after transition to heterotrophic condition.

other, producing the response to heterotrophic condition. Comparison of GO for DAS and DEG sets shows that the frequency of GO terms is different (Fig 3).

Genes categorized by 'binding' and 'response to stimuli', for example, were higher in DAS than DEG set. Meanwhile, 'oxidoreductase activity', 'transcription regulatory activity', and 'cofactor binding' were highly enriched in DEGs. In molecular function category, binding (GO: 1901363), protein binding (GO: 0005515), catalytic activity (GO: 0003824), molecular function regulator (GO: 0098772), antioxidant activity (GO: 0016209), molecular transducer activity (GO: 0060089), molecular transducer activity (GO: 0060089) and transcription regulator activity (GO: 0140110) were significantly enriched in DAS genes (P-value < 0.05). In the GO term binding (GO: 0005488), heterocyclic compound binding (GO: 1901363) including nucleic acid binding (GO: 0003676), vitamin B6 binding (GO: 0070279) and tetrapyrrole binding (GO: 0046906), protein binding (GO: 0005515) including protein dimerization activity (GO: 0046983), unfolded protein binding (GO: 0051082), heat shock protein binding (GO: 0031072), transcription factor binding (GO: 0008134) were significant.

## Alternative splicing and expression patterns of the carotenoids biosynthesis related genes

We analyzed the genes involved in carotenoid biosynthesis. Ten *A. protothecoides* genes that encode enzymes in the carotenoid biosynthetic pathways, including geranylgeranyl

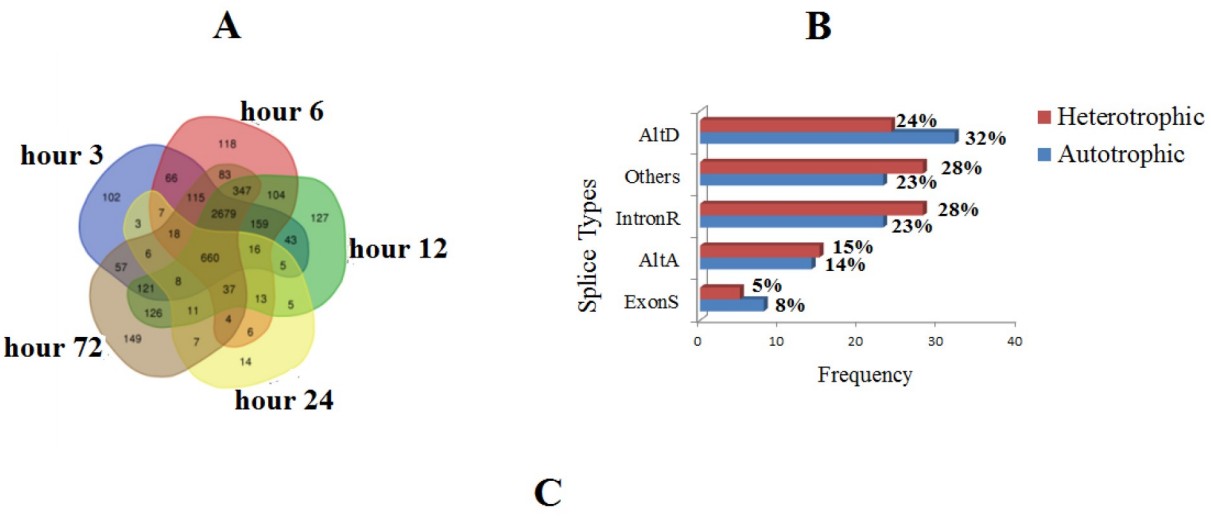

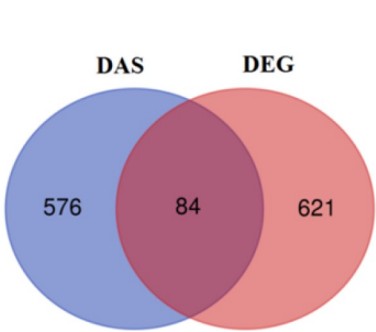

**Fig 2.** Profiles of DAS in microalgae after transition to heterotrophic condition, overlap of DAS genes at five time points (A), catalogue of AS event in autotrophic and heterotrophic condition (B) and overlap of DAS and DEGs (C) in heterotrophic condition.

pyrophosphate synthase 1(GGPS1), phytoene synthase (PSY), phytoene desaturase (PDS), ζ-carotene desaturase (ZDS), lycopene epsilon cyclase (LCYE), lycopene beta cyclase (LCYB1), β-carotene hydroxylase (CHYB), P450 hydroxylases (P450-CHY), Carotenoid isomerase (CrtISO), and zeaxanthin epoxidase (ZEP) were selected for alternative splicing and expression analysis. The heat map of the carotenoid biosynthesis related genes was drawn according to fold change values (Fig 4).

Except the P450b-CHY, ZDS, and LCYB1, expression of the most carotenoid biosynthetic pathway genes decreased at heterotrophic growth. We next tried to investigate whether heterotrophic growth condition would induce alternative splicing of carotenoids biosynthesis related genes. Results indicated that heterotrophic condition affects AS pattern of carotenoids biosynthesis related genes. Comparison of AS pattern in autotrophic and heterotrophic growth condition showed that PDS, LCYB1, CrtISO, ZEP, and LCYE undergo AS process at autotrophic condition, whereas in heterotrophic condition, AS events were detected for the PDS, P450-CHY, ZDS, and CrtISO, highlighting the regulatory impacts of growth condition changes on splicing pattern of carotenoids biosynthesis pathway.

## Discussions

Recently developed high throughput sequencing technology so called Next Generation Sequencing (NGS) has allowed effective genome-wide detection of alternative splicing events.

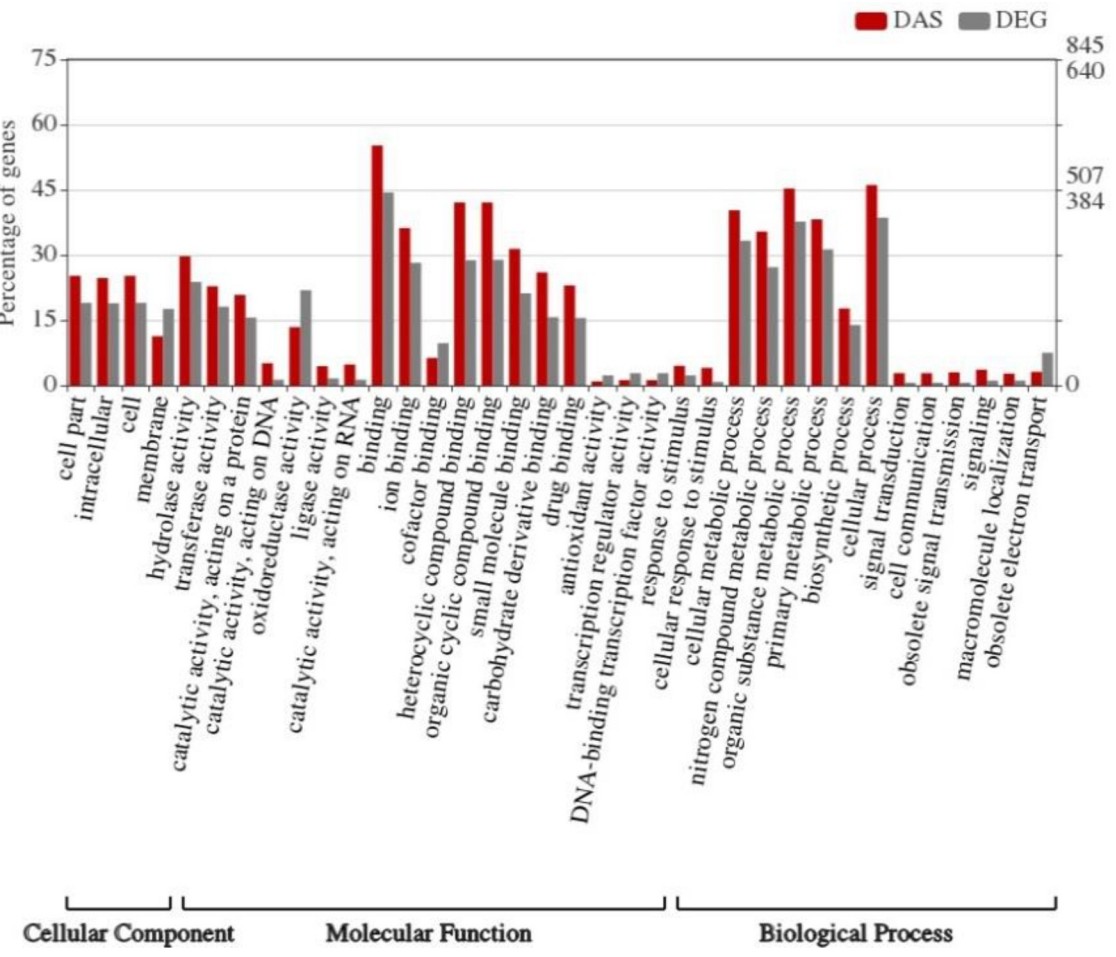

**Fig 3. Functional enrichment analysis of DEG and DAS genes.** Significantly (P-value<0.05) enriched gene ontology in three categories including cellular component, molecular condition and biological process are shown in below of the figure.

In current study, we used RNA-seq data to uncover features of alternative splicing and gene expression in *A.protothecoides* grown in heterotrophic condition. Comparison of DEG and DAS showed that it was relatively small overlap between DAS and DEG genes (84 genes) indicated that AS accomplish as a distinct adaptive regulatory layer in heterotrophic condition. Functional enrichment analysis revealed that the DAS genes were involved mainly in several biological processes, including cellular metabolic process, nitrogen compound metabolic process, and biosynthetic process. Further analysis revealed that DAS genes were involved tetrapyrrole binding (GO: 0046906). Among different genes involved in tetrapyrrole binding, the gene encodes the multidrug resistance (MDR)-like ABC transporters undergoes AS events in heterotrophic condition. This protein has been shown to catalyze the primary active export of auxin [23] highlighting the functional importance of AS event in signaling rewiring during growth mode change. Among different stress related genes, DnaJ-like proteins belong to heat shock protein 40 (Hsp40) families were undergoing AS events in heterotrophic condition. It has been demonstrated that DnaJ-like proteins participate in regulation of protein folding, protein transport and cellular responses to stress [24]. Intriguingly, transcription regulator activity significantly enriched in DAS genes. More dissection of DAS genes that were categorized in transcription regulator activity, showed that DNA-binding transcription factor

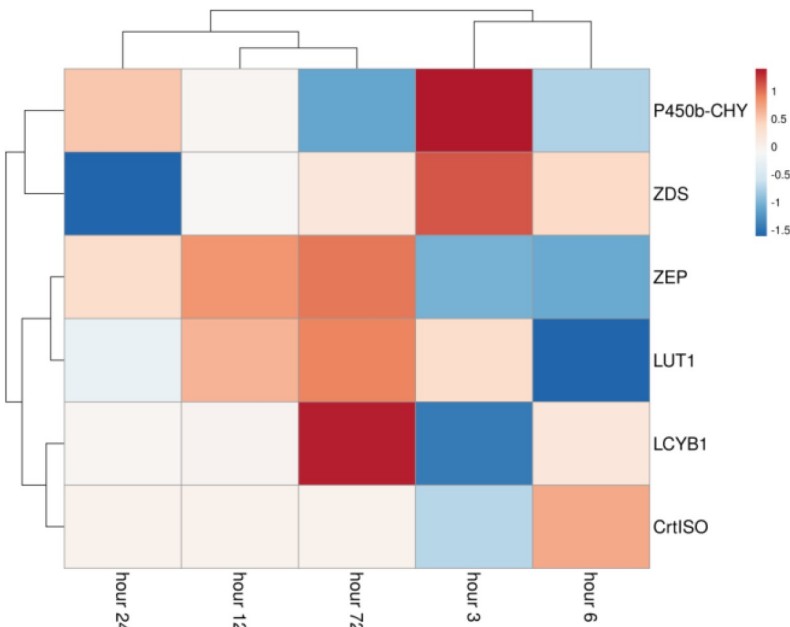

**Fig 4. Expression profile of carotenoid biosynthesis related genes after transition to heterotrophic condition.**

activity, transcription initiation factor activity are mostly enriched in DAS genes, whereas these categories were not found in DEGs which indicated that AS can serve as an independent mechanism for gene regulation in response to growth mode change [25, 26].

The most striking result to emerge from the data is that growth in heterotrophic condition affects AS pattern of carotenoids biosynthesis pathway. Among different genes contributed in carotenoids biosynthesis pathway, PDS, P450-CHY, ZDS and CrtISO underwent AS event. AltD, AltA, IntronR, and AltD events were detected for PDS, P450-CHY, ZDS, CrtISO, respectively. More interestingly, similar pattern of AS were also observed in training data set for carotenoids biosynthesis related genes.

PDS, a key enzyme in carotenoid biosynthesis, are needed in the conversion from phytoene to lycopene [27]. It has 8 exons (E) of which 3 exons show significant DEU and 5 exons are conserved usage at heterotrophic condition in compared with autotrophic condition. E3 and E8 of PDS exhibit time point-specific exon usage at heterotrophic condition. Only E1 and E2 in this gene were consistently showed DEU at all-time points (Fig 5). In agreement with our finding, it has been previously suggested that alternative transcripts could regulate the cellular concentrations of PDS, depending on the physiological or environmental conditions [28].

ZDS and CrtISO convert ζcarotene to lycopene, which is the substrate for two competing lycopene cyclases, ε-LCY and β-LCY [29]. It has been reported that overexpression of ZDS significantly increase the lutein content and salt tolerance [30]. Finding of our study showed that ZDS undergoes AS event (IntronR) at heterotrophic growth condition (time point 6, 12, and 72 h). ZDS has ten exons of which three exons showed significant DEU at heterotrophic condition. Exon 4 and 7 consistently showed DEU at hour 6, 12, and 72, after transition to the heterotrophic condition, however, DEU of exon 10 was only found at hour 72.

Regarding the CrtISO, the number of exons with DEU was varied from hour 3 to 72. It is apparent from the results that at 3 and 6 hour after transition to the heterotrophic condition, E 5 and E 11 show DEU, it is whilst at hour 12 and 72 the number of exons with DEU increased to 4 (E1, E2, E7, and E 11) and 5 (E1, E2, E5, E7 and E11), respectively. Whereas no significant

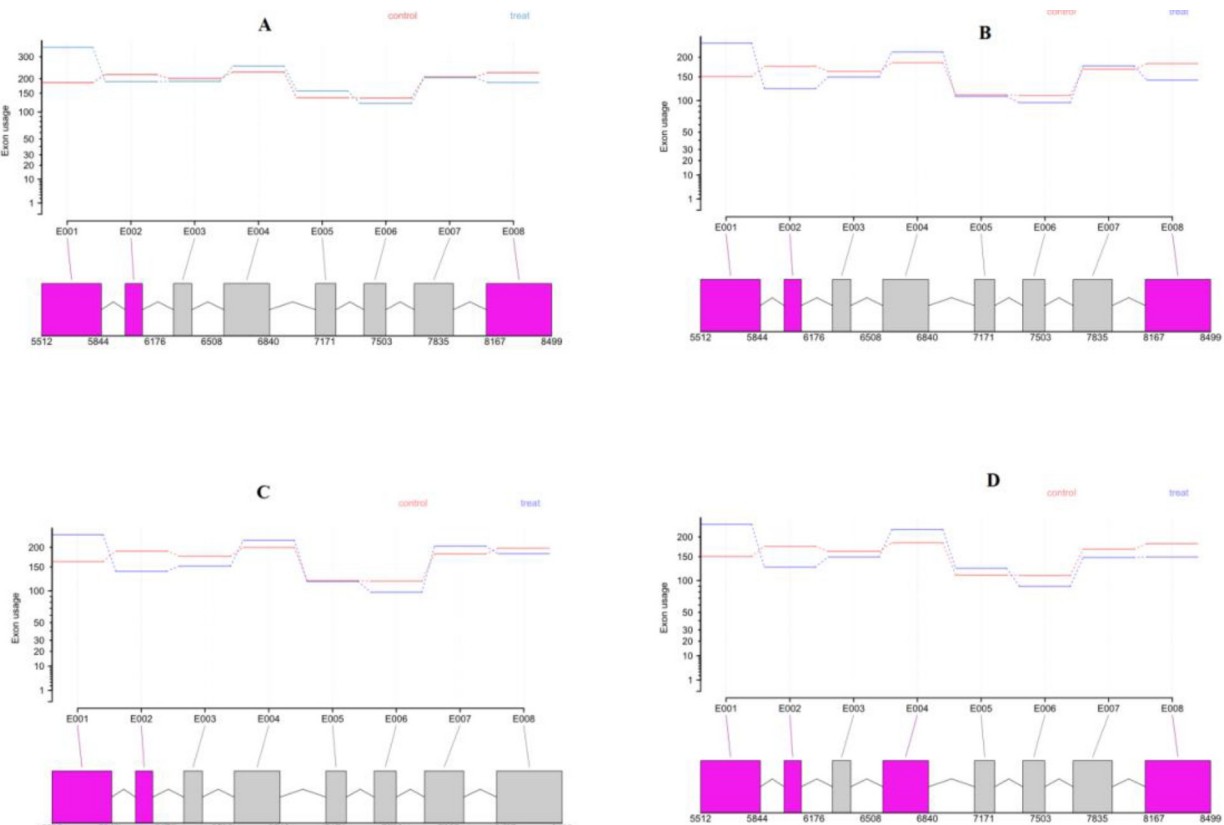

**Fig 5. Gene models illustrating AS patterns of PDS gene.** Exon usage was used to measure exon expression after normalizing for total gene expression biases. Significant differentially expressed exons are colored in magenta (p.adjust<0.05) in 6h (A), 12h (B), 24h (C) and 48h (D) after transition to heterotrophic condition.

expression was identified at hours 24 and 72, highlighting the importance of post-transcriptional modification processes on carotenoids biosynthesis pathway regulation during adaptation to different growth condition. In line with our hypothesis, it has been suggested that the translational control mechanisms is interpreted in terms of flux adjustments needed in response to retrograde signals stemming from intermediates of the plastid-localized carotenoid biosynthesis pathway [31].

## Conclusions

In current study, we present a RNA-seq analysis of a comparison of the transcriptional and post-transcriptional responses of *A. prototheoides* grown under autotrophic and heterotrophic conditions. More AS events in the heterotrophic condition were identified than in the autotrophic condition. There is relatively small overlap between DAS and DEG gene indicated that AS accomplish as a distinct adaptive regulatory layer in responses to growth condition change. To the best of our knowledge, this is the first study in which alternative splicing has investigated in the microalgae grown on heterotrophic and autotrophic conditions. The results of current study provides new insight into underlying regulatory responses of microalgae grown in modified condition to optimized production of high value metabolites at industrial scale.

## Supporting information

**S1 Table. Alternative splicing landscape analysis in autotrophic condition.**
(TXT)

**S2 Table. Alternative splicing landscape analysis in heterotrophic condition.**
(GTF)

**S3 Table. Differentially expressed genes and their corresponding fold changes at 6, 12, 24, and 72 hour after transition to heterotrophic condition growth in heterotrophic condition.**
(XLSX)

**S4 Table. Differentially spliced genes at 6, 12, 24, and 72 hour after transition to heterotrophic condition growth in heterotrophic condition.**
(XLSX)

## Acknowledgments

We would like to thank Dr Asgar Panahi for reading of the manuscript.

## Author Contributions

**Conceptualization:** Bahman Panahi.

**Data curation:** Bahman Panahi.

**Formal analysis:** Bahman Panahi.

**Validation:** Bahman Panahi.

**Writing – original draft:** Bahman Panahi, Mohammad Amin Hejazi.

**Writing – review & editing:** Bahman Panahi, Mohammad Amin Hejazi.

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
