## [Decision Letter · Decision Letter 0]

20 Apr 2020

PONE-D-20-05724

Integrated analysis of differential expression and alternative splicing in microalgae grown under heterotrophic condition

PLOS ONE

Dear Dr Panahi,

Thank you for submitting your manuscript to PLOS ONE. After careful consideration, we feel that it has merit but does not fully meet PLOS ONE’s publication criteria as it currently stands. Therefore, we invite you to submit a revised version of the manuscript that addresses the points raised during the review process.

It is a simple bioinformatic study to compare the data of transcriptional and post-transcriptional RNA. Study will be important if it could have been supported lab experimentation.  Authors stated, “ The results also indicated that prevalence of Intron retention is associated with down-regulation of the genes involved in carotenoid biosynthesis”. This needs to be supported with the practical experimentation to validate your statement and make study useful to wide readership of Plos-One. Results and Discussion parts are not clearly written. I would appreciate if you could sperate them and write more lucidly. Please state, how your current study differed from “Systems biology approach identifies functional modules and regulatory hubs related to secondary metabolites accumulation after transition from autotrophic to heterotrophic growth condition in microalgae” that published recently.

We would appreciate receiving your revised manuscript by within six months. To enhance the reproducibility of your results, we recommend that if applicable you deposit your laboratory protocols in protocols.io, where a protocol can be assigned its own identifier (DOI) such that it can be cited independently in the future. For instructions see: http://journals.plos.org/plosone/s/submission-guidelines#loc-laboratory-protocols

We look forward to receiving your revised manuscript.

Kind regards,

Shashi Kumar, Ph.D.

Academic Editor

PLOS ONE

Additional Editor Comments:

It is a simple bioinformatic study to compare the data of transcriptional and post-transcriptional RNA. Study will be important if it could have been supported lab experimentation. Authors stated, “ The results also indicated that prevalence of Intron retention is associated with down-regulation of the genes involved in carotenoid biosynthesis”. This needs to be supported with the practical experimentation to validate your statement and make study useful to wide readership of Plos-One. Results and Discussion parts are not clearly written. I would appreciate if you could sperate them and write more lucidly. Please state, how your current study differed from “Systems biology approach identifies functional modules and regulatory hubs related to secondary metabolites accumulation after transition from autotrophic to heterotrophic growth condition in microalgae” that published recently.

'The authors would like to thank Agricultural Biotechnology Research Institute of Iran for

financial support.'

'No'

Please clarify the sources of funding (financial or material support) for your study. List the grants or organizations that supported your study, including funding received from your institution.State what role the funders took in the study. If the funders had no role in your study, please state: “The funders had no role in study design, data collection and analysis, decision to publish, or preparation of the manuscript.”If any authors received a salary from any of your funders, please state which authors and which funders.If you did not receive any funding for this study, please state: “The authors received no specific funding for this work.”Please include your amended statements within your cover letter; we will change the online submission form on your behalf.

3. Please upload a copy of Figure 6, to which you refer in your text. If the figure is no longer to be included as part of the submission please remove all reference to it within the text.

5. Please include captions for your Supporting Information files at the end of your manuscript, and update any in-text citations to match accordingly. Please see our Supporting Information guidelines for more information: http://journals.plos.org/plosone/s/supporting-information

Reviewers' comments:

Reviewer's Responses to Questions

**Comments to the Author**

1. Is the manuscript technically sound, and do the data support the conclusions?

Reviewer #1: Yes

Reviewer #2: Partly

2. Has the statistical analysis been performed appropriately and rigorously? 

Reviewer #1: Yes

Reviewer #2: Yes

3. Have the authors made all data underlying the findings in their manuscript fully available?

Reviewer #1: Yes

Reviewer #2: No

4. Is the manuscript presented in an intelligible fashion and written in standard English?

Reviewer #1: Yes

Reviewer #2: Yes

5. Review Comments to the Author

Reviewer #1: The article is a well-written, needed, and helpful to understand the relationship between growth conditions and differential gene expression pattern. The interesting thing that I liked is how alternative splicing events can change the mode of metabolism in response to growth condition change. I thought this article and previous article about identification of functional modules and regulatory hubs may help us to study the principal factors that can change specific metabolism to get specific product for commercial production.

1. Kindly correct the word “evenets”. ( Abstract-line No.5)

Reviewer #2: Summary:

This paper is an analysis of the transcriptional and post transcriptional responses of algae grown under heterotrophic conditions for the production of high value metabolites. The ability of algae to reduce the nutrients in the environment while still extracting a usable product is an important factor in moving towards energy sustainability and effective nutrient management in an algal cultivation. In order to efficiently proceed with the growth of algae for industrial productions under controlled conditions, understanding the influence of different nutrient quality parameters and nutrients is important. This paper brings a light forward knowledge supporting this understanding.

Overall Impression:

This work appears to bring new knowledge on the transcriptional and post-transcriptional responses of alga towards the growth conditions influenced by nutrient inputs. However, the details of nutrient parameters in two conditions are not stated by the authors and also how they are influencing above process. It should require more clarifications for claiming their suitability to novelty.

General Editing comments:

The paper requires a detailed round of editing. There are some examples of missing legends and references in the text. The user should be able to read the caption and have all the information required to understand the figure or table.

Specific Comments:

a) Add a brief description of the autotrophic and heterotrophic culture conditions and nutrient parameters used in this study. Because this work emphasizes the growth conditions and it will help in understanding the context.

b)Mention the correct test and control sample size and their replications; ie. In the material and methods section Page No. 9 first paragraphs (The sample size of autotrophic growth conditions and their test timings).

c)Page number 11, figure 2B: The data presented here are quite interesting. However, authors need to make a connection between these data with the possible importance/influence of the algal metabolite production.

d)In the result section, functional impact of DEG and DAS genes, in third paragraph. The authors state that “Transcription regulator activities of DEGs are significant …” Are these statistical differences observed after the analytical experiment? If so, where are the stats to prove the statements in this paragraph? If a statistical analysis was completed, please add the test statistic and p values either embedded in the text or in a graph figure legend.

e)Figure legend ‘C’ is present in the text, but absent in all mentioned figures.

f)Page No. 8 reference ‘Huang & Xu, 2015’ is present in the text, but absent in the reference list

g)Page No. 15 line number 13th journal description is present in the reference section but author detail is absent in text as well as in reference.

h)Page No. 15 Kong et al. Journal page number missing

i)Page No. 15 Panahi et al., typographic error in year.

j)Page No. 16 Ravichandran et al. and Xiao et al. References journal page number missing.

k)Page No. 16 Panahi et al., typographic error in journal page number

l)Page No. 16 Schurch & Schofield (2016) Journal name not mentioned

m)Reference journal name consistency; either follow full name/abbreviation/italic pattern/non italic.

6. PLOS authors have the option to publish the peer review history of their article (what does this mean?). If published, this will include your full peer review and any attached files.

Reviewer #1: No

Reviewer #2: No

---

## [Author Response · Author response to Decision Letter 0]

24 Apr 2020

Reviewer comments

Response to Editor comments

Dear editor thank you for your useful comments.

 we revised the manuscript thoroughly revised. As your comment we separated results and dissuasion sections and revised these sections. Regarding the validation with wet lab experiments, unfortunately Auxenochlorella protothecoides is not available for our institution and also contrary because of limitation of biological material transformation. Therefore, we had to validate the results with another RNA seq data sets follow the procedure that has been proposed by Feng et al. 2018. Characterization of kinase gene expression and splicing profile in prostate cancer with RNA-Seq data. BMC Genomics. 

Regarding the differences with our previous published manuscript, in our previous manuscript, we focused on connectivity analysis and functional module detection in network analysis, where as in current study we analysis gene expression and alternative splicing with comparative approach.

Reviewer #1: 

Kindly correct the word “evenets”. ( Abstract-line No.5) 

Response: It was revised

Reviewer #2:

This work appears to bring new knowledge on the transcriptional and post-transcriptional responses of alga towards the growth conditions influenced by nutrient inputs. 

However, the details of nutrient parameters in two conditions are not stated by the authors and also how they are influencing above process.

It was revised

It should require more clarifications for claiming their suitability to novelty.

General Editing comments:

The paper requires a detailed round of editing. There are some examples of missing legends and references in the text. The user should be able to read the caption and have all the information required to understand the figure or table.

It was revised

Specific Comments:

a) Add a brief description of the autotrophic and heterotrophic culture conditions and nutrient parameters used in this study. Because this work emphasizes the growth conditions and it will help in understanding the context.

It was revised

b) Mention the correct test and control sample size and their replications; ie. In the material and methods section Page No. 9 first paragraphs (The sample size of autotrophic growth conditions and their test timings).

It was revised

c)Page number 11, figure 2B: The data presented here are quite interesting. However, authors need to make a connection between these data with the possible importance/influence of the algal metabolite production.

Dear reviewer in the end of the “discussion” section we discussed this relationships

d) In the result section, functional impact of DEG and DAS genes, in third paragraph. The authors state that “Transcription regulator activities of DEGs are significant …” Are these statistical differences observed after the analytical experiment? If so, where are the stats to prove the statements in this paragraph? If a statistical analysis was completed, please add the test statistic and p values either embedded in the text or in a graph figure legend.

Dear reviewer enrichment analysis thoroughly tested based on statically methods implemented in WEGO. we included (P-Value<0.05) to show significance of enriched ontology in the text and figure legend. 

e)Figure legend ‘C’ is present in the text, but absent in all mentioned figures.

It was revised

f)Page No. 8 reference ‘Huang & Xu, 2015’ is present in the text, but absent in the reference list

It was revised

g)Page No. 15 line number 13th journal description is present in the reference section but author detail is absent in text as well as in reference.

It was revised

h)Page No. 15 Kong et al. Journal page number missing

It was revised

i)Page No. 15 Panahi et al., typographic error in year.

It was revised

j)Page No. 16 Ravichandran et al. and Xiao et al. References journal page number missing.

It was revised

k)Page No. 16 Panahi et al., typographic error in journal page number

It was revised

l)Page No. 16 Schurch & Schofield (2016) Journal name not mentioned

It was revised

m)Reference journal name consistency; either follow full name/abbreviation/italic pattern/non italic.

Dear reviewer it was revised

---

## [Decision Letter · Decision Letter 1]

2 Jun 2020

Integrative analysis of gene expression and alternative splicing in microalgae grown under heterotrophic condition

PONE-D-20-05724R1

Dear Dr. Panahi,

We are pleased to inform you that your manuscript has been judged scientifically suitable for publication and will be formally accepted for publication once it complies with all outstanding technical requirements.

With kind regards,

Shashi Kumar, Ph.D.

Academic Editor

PLOS ONE

Additional Editor Comments (optional):

Thanks for making all corrections as suggested by reviewers.

Reviewers' comments:

Reviewer's Responses to Questions

**Comments to the Author**

1. If the authors have adequately addressed your comments raised in a previous round of review and you feel that this manuscript is now acceptable for publication, you may indicate that here to bypass the “Comments to the Author” section, enter your conflict of interest statement in the “Confidential to Editor” section, and submit your "Accept" recommendation.

Reviewer #2: All comments have been addressed

2. Is the manuscript technically sound, and do the data support the conclusions?

Reviewer #2: Yes

3. Has the statistical analysis been performed appropriately and rigorously? 

Reviewer #2: Yes

4. Have the authors made all data underlying the findings in their manuscript fully available?

Reviewer #2: Yes

5. Is the manuscript presented in an intelligible fashion and written in standard English?

Reviewer #2: Yes

6. Review Comments to the Author

Reviewer #2: (No Response)

7. PLOS authors have the option to publish the peer review history of their article (what does this mean?). If published, this will include your full peer review and any attached files.

Reviewer #2: No

---

## [Editor Report · Acceptance letter]

8 Jun 2020

PONE-D-20-05724R1 

Integrative analysis of gene expression and alternative splicing in microalgae grown under heterotrophic condition 

Dear Dr. Panahi:

I'm pleased to inform you that your manuscript has been deemed suitable for publication in PLOS ONE. Congratulations! Your manuscript is now with our production department. 

Kind regards, 

on behalf of

Dr. Shashi Kumar 

Academic Editor

PLOS ONE